# Taxonomy, Habitat Preference, and Niche Overlap of Two Arrow-Poison Flea Beetle Species of the Genus *Polyclada* in Sub-Saharan Africa (Coleoptera, Chrysomelidae)

**DOI:** 10.3390/insects13080668

**Published:** 2022-07-23

**Authors:** Maurizio Biondi, Paola D’Alessandro, Francesco Cerasoli, Walter De Simone, Mattia Iannella

**Affiliations:** Department of Life, Health & Environmental Sciences, University of L’Aquila, Via Vetoio-Coppito, 67100 L’Aquila, Italy; maurizio.biondi@univaq.it (M.B.); francesco.cerasoli@univaq.it (F.C.); walter.desimone@graduate.univaq.it (W.D.S.); mattia.iannella@univaq.it (M.I.)

**Keywords:** Galerucinae Alticini, Afrotropical region, new synonymy, lectotype designations, aedeagal and spermathecal shapes, ecological niche modelling, GIS analysis

## Abstract

**Simple Summary:**

The taxonomy of many African Coleoptera species is remains poorly known, and the knowledge of their ecological requirements is worse still. Starting with original data, we describe morphological differences and ecological data for two flea beetle species, *Polyclada bohemani* and *P. pectinicornis*, which traditionally have been used by the Bushmen people in sub-Saharan Africa to poison their arrows. Moreover, we evidence differences in the formations of vegetation used by these two species, which are known to occur mainly in savannah and open forest habitats. Also, we identify differently suitable areas in terms of climatic preferences, in addition to a common territory in East Africa. We also supply, for the first time, the description of the shape of the aedeagus and the spermatheca of both species, supplying important new diagnostic characters for their identification.

**Abstract:**

Coupling the geographic distribution and the ecological requirements of species often supports taxonomy and biogeography. In this contribution, we update the distribution of two flea beetle species of ethno-entomological interest, *Polyclada bohemani* and *P. pectinicornis*, by analyzing original data. In addition, we supply their main morphological diagnostic characters, describing their aedeagal and spermathecal shapes for the first time. We also assess their niche differences in terms of climatic and vegetation needs, by means of ecological niche modelling and remote sensing techniques. Several new localities were identified to improve knowledge of the geographical distribution of both species. Moreover, we located a wide climatic suitability overlap in East Africa for these two flea beetle species, while in other areas they show a clear separation. Our analysis also reports that *P. bohemani* is associated with areas of denser tree cover than *P. pectinicornis*. Finally, the lectotypes of *Diamphidia bohemani* Baly, 1861, *Clytra pectinicornis* Olivier, 1791, and *Diamphidia compacta* Fairmaire, 1887 are here designated and the new synonymy *Clytra pectinicornis* Olivier = *Diamphidia compacta* Fairmaire syn.nov. is proposed.

## 1. Introduction

The Coleoptera, with about 380,000 described species, are the world’s most diverse animal order. Their truly amazing morphological and functional solutions translate into critical ecological functions, ranging from plant pollination to nutrient recycling in various environmental matrices (e.g., soil, freshwaters), to secondary consumer food supply and pest biocontrol [1,2,3]. Despite their primary significance in many environments, the ecological and biogeographical variables that affect their distribution in many areas, especially in the southern hemisphere, have yet to be understood.

The family Chrysomelidae, known as leaf beetles, is among the most frequently represented of the Coleoptera families in the terrestrial habitats of all continents, except the polar regions. Chrysomelidae includes over 40,000 described species, across more than 2330 genera and 12 subfamilies (Bruchinae, Cassidinae, Eumolpinae, Lamprosomatinae, Cryptocephalinae, Criocerinae, Chrysomelinae, Galerucinae, Donaciinae, Sagrinae, Spilopyrinae and Synetinae) [4]. In particular, the Alticini, included in the subfamily Galerucinae, comprise the largest and most diverse tribe of leaf beetles, with over 540 genera and about 8000 extant species [5], and occur all over the world. Alticini are commonly defined as “flea beetles” because they have a metafemoral extensor tendon that enables them to jump [6]. Adult and larval stages feed mainly on the stems, leaves, or roots, but rarely on the flowers, of almost all the higher plant families in different environments, being sometimes classified as pests [7,8,9] and generally showing high levels of specialization [10,11,12]. The Afrotropical flea beetle fauna includes about 1600 known species in 103 genera; among them, the genus *Polyclada* Chevrolat widely occurs in sub-Saharan Africa and the Arabian Peninsula [13,14,15]. Sixteen species are currently attributed to the genus, although at least four of these actually belong to different genera, while others await description (Biondi, unpublished data). *Polyclada* belongs to the *Blepharida* group of genera sensu Furth (1998) [16], and Prathapan and Chaboo (2011) [17], along with *Diamphidia* Gerstaecker, *Xanthophysca* Fairmaire, and the recently re-evaluated *Blepharidina* Bechyné and *Calotheca* Heyden [11,15,18,19,20,21,22]. *Polyclada* species are associated with the Anacardiaceae (*Sclerocarya birrea*) and Burseraceae (*Commiphora* spp.) plant families, which are found in a variety of woodland and savannah ecosystems [22,23].

To explore and define environment–occurrence trends in the Afrotropical region for two species of this flea beetle genus, *Polyclada bohemani* (Baly) and *P. pectinicornis* (Olivier), we used state-of-the-art ecological niche models (ENMs), together with habitat and niche overlap analyses. Both species are well-known for their ethno-entomological importance, as the pupae and body fluids are employed as arrow poison by the San people (Bushmen) of southern Africa. The active molecule is the diamphotoxin, present also in the African flea beetle *Diamphidia* Gerstaecker, a toxalbumin that works similarly to various snake venoms, causing widespread paralysis, hemolysis, and death (cf. Chaboo et al., 2007).

In the present contribution, in addition to updated distributions, we also provide the main diagnostic characters for the identification of these two African species, reporting for the first time the description of the median lobe of the aedeagus and of the spermatheca. Moreover, we characterize their ecological requirements and interpret their pattern of occurrence at a continental scale.

## 2. Materials and Methods

### 2.1. Study Area, Species Database, and Vegetation Formations

The study area comprised sub-Saharan Africa. We performed analyses on a dataset including occurrences for two endemic flea beetle species (Chrysomelidae, Galerucinae, Alticini) of the genus *Polyclada*, *P. pectinicornis* (Olivier), with 80 occurrence localities, and *P. bohemani* (Baly), with 104. We retrieved data records of the two target species mainly from the examined material, consisting of 550 dried pinned specimens preserved in the institutions listed in the “Abbreviations Used” section below. The specimens were examined and dissected using a Leica M205C stereomicroscope. Photographs of the habitus and the spermatheca were taken using a Leica DMC5400 camera and compiled using Zerene Stacker software v. 1.04. Scanning electron micrographs were taken using a Hitachi TM-1000. Terminology follows D’Alessandro et al. (2016) [24] for the median lobe of the aedeagus and spermatheca. We recorded the geographic coordinates for the localities in decimal degree system using the WGS84 datum, and we added information included in square brackets to the label data, using the Google Earth website for coordinates and geographic information. Abbreviations for the depositories follow the list on the website The Insect and Spider Collections of the World [25]. Chorotypes follow Biondi and D’Alessandro (2006) [26]. We used the raster map of terrestrial ecosystems of Africa to gather spatial information (resolution: 90 m cell) about vegetation formations, classified in a hierarchical fashion (i.e., class, subclass, formation, division, macro-group) [27]. 

Moreover, we assessed tree cover within the habitats in which the two species occur. To assess this environmental predictor, we extracted the values of tree cover (%) from the dataset of Hansen et al. (2013) (updated to 2021) at a spatial resolution of 30 m. Tree canopy cover was defined as canopy closure for all vegetation taller than 5 m. Based on Landsat images, this data was acquired by the Google Earth Engine cloud platform using a Python-based code, which allowed us to extract the percentage value of tree cover for each occurrence of the two target species.

### 2.2. Model Building

To estimate the current suitable areas for *Polyclada bohemani* and *P. pectinicornis*, we built ecological niche models. For this purpose, we downloaded 19 temperature- and precipitation-related “bioclimatic” raster variables from the Worldclim 2.1 online repository [28] at cell resolution of 2.5 arcminutes (~5 km at the equator). To avoid possible correlation among predictors, which might lower the model’s performance, we assessed both the variance inflation factor (VIF, threshold set = 10 following Guisan et al. (2017)) [29], and Pearson’s r (|r| < 0.9, following Dormann (2007) [30] and Elith et al. (2006)) [31], using the ‘vifstep’ and ‘vifcor’ functions of the ‘usdm’ R package [32], then applying a subset of selected predictors to calibrate the models. 

The building of the ENMs for both *Polyclada* species was completed through the “biomod2” package [33] in R environment [34]. Ensemble models (EMs, resulting from the combination of individual ENMs) were built for the current climatic conditions using the “BIOMOD_EnsembleModeling” function. We generated 10 sets of 1000 pseudo-absences using the Surface Range Envelope method [35,36], with quantile = 0.05. We parametrized models built for *P. bohemani* and *P. pectinicornis* as follows: generalized linear models (GLM): type = “quadratic”, interaction level = 3; multiple adaptive regression splines (MARS): type = “quadratic”, interaction level = 3; generalized boosting model, also known as boosted regression trees (BRT): number of trees = 5000, interaction depth = 3, cross-validation folds = 10; we performed five evaluation runs. 

### 2.3. Model Evaluation and Ensemble Forecast

To assess the discrimination performance of the single models, we took advantage of two different evaluation metrics, the area under the curve (AUC) of the ROC curve [37] and the True Skill Statistics (TSS) [38]. We used 80% of the initial dataset to build the models, and the remaining 20% for their validation. Considering the five evaluations runs, 10 pseudo-absences sets, and three modeling algorithms chosen, 150 single models were finally generated for each species. We then built the EMs by selecting only the ENMs exceeding the thresholds TSS > 0.7 and AUC > 0.7; the “weighted mean of probabilities” (wmean), which averages the single models by weighting their AUC or TSS scores, was used for this purpose [33].

### 2.4. Environmental Niche Overlap

The extent of possible environmental niche overlap between *Polyclada bohemani* and *P. pectinicornis* was investigated through the PCA-Env approach [39], taking advantage of the ‘ecospat’ R package version 3.1 [40]. A principal component analysis (PCA) was calibrated by sampling the values of the considered covariates from occurrence localities of the target species, as well as from several background points across the study area. Subsequently, a two-dimensional gridded environmental space was defined based on the two principal components (hereafter, PrinComp) contributing the most to explained variance (i.e., showing the highest eigenvalues). Finally, ‘niche occupancy’ of each target species within this PCA-derived 2-D space was computed as a kernel-smoothed density of occurrence, correcting the observed density of occurrence by the density of environmental conditions possibly exploitable by the species (i.e., those within reachable areas). To calibrate the PCA-Env, we coupled a subset of uncorrelated bioclimatic variables (found by means of the VIF analysis described above), with cell-by-cell percentage cover of three vegetation formations, namely tropical lowland humid forest (1.A.2), tropical lowland grassland, savanna and shrubland (2.A.1), and warm desert and semi-desert scrub and grassland (3.A.2) [27]. The chosen formations were those contributing most to the first two PrinComps of a preliminary PCA-Env, calibrated using only percentage cover of the seventeen African vegetation formations as covariates. Niche overlap between *P. bohemani* and *P. pectinicornis* was computed using Schoener’s *D* metric and a modified version of Hellinger’s distance metric, indicated as *I* [41]. The obtained niche overlap scores were tested for significant deviation from null expectation, considering possible differences between species in terms of exploitable environments, through the niche similarity test implemented in ‘ecospat’. In this test, the observed overlap score between two species, for exaqmple *j* and *k*, is compared to a distribution of simulated overlap scores; such a distribution is derived by comparing, *n* times, the simulated niche occupancies of *j* and *k*, where simulated occupancies are defined by randomly sampling environmental conditions from a background area around the occurrence localities of each species. In this manner, environmental niches of the two species can be considered more or less similar than expected, based on differences between species in exploitable environments, if the observed overlap of the simulated scores is higher than the 95th percentile (i.e., niche conservatism) or lower than the 5th percentile (i.e., niche divergence), respectively [39]. Based on the limited dispersal capacity of *P. bohemani* and *P. pectinicornis*, we defined the respective background areas by drawing a circular buffer of 20 km around their occurrences. The distribution of simulated overlap scores was obtained by repeating the occupancy simulation process 250 times.

### 2.5. Abbreviations Used

Collections and depositories. BAQ: Italy, University of L’Aquila, Collection of M. Biondi; MNHN: France, Paris, Muséum National d’Histoire Naturelle; NHMUK: United Kingdom, London, Natural History Museum; NMPC: Czech Republic, Prague, National Museum (Natural History); SANC: South Africa, Pretoria, South African National Collection of Insects; ZSM: Germany, München, Zoologische Staatssammlung. 

Biometrics. LA: numerical sequence from base to apex proportional to the length of each antennomere; LAED: length of aedeagus; LAN: length of antennae; LB: total body length (from apical margin of head to apex of elytra); LE: length of elytra; LP: medial length of pronotum; LSP: maximum length of spermatheca, including ductus; WE: maximum width of elytra combined; WP: maximum width of pronotum; (!): new record. 

## 3. Results

### 3.1. Taxonomy

*Polyclada bohemani* (Baly)

*Diamphidia bohemani* Baly, 1861: 198–199 [42]. 

*Cladotelia bohemani* (Baly): Kolbe 1897 [43]: 335 [43]; Weise 1902: 162 [44]. 

*Cladocera bohemani* (Baly): Ferreira 1963: 519 [45].

*Polyclada bohemani* (Baly): Bechyné 1957 [46]: 182; Scherer 1962: 78 [47]; Scherer 1972: 16 [48]; Chaboo 2012: 70–71 [49].

Locus typicus: “Port Natal”.

Type material: Lectotype ♂: “*Diamphidia Bohemani*/Baly/Port Natal/Type//Baly Coll.//Type//NHMUK014380611” (NHMUK) (here designed by M. Biondi and P. D’Alessandro). 

Distribution: Angola (!), Botswana (!), Democratic Republic of Congo, Kenya, Malawi (!), Mali (!), Mozambique, Namibia (!), Nigeria (!), Republic of South Africa, Somalia (!), South Sudan, Tanzania, Zambia (!), and Zimbabwe. Chorotype: Afrotropical (AFT).

Localities examined: Angola: Cunene prov., 20 kms N of Kahama, 26–27.xi.2013 [−3.657 Lat, 32.595 Lon]. Botswana: Gaborone, 23.xi.2020, −24.677 Lat, 25.879 Lon (from iNaturalist); ditto, 15.xii.2018 (from iNaturalist). Democratic Republic of Congo: Lindi River [1.227 Lat, 25.557 Lon]; Lualaba, Kapanga [−10.478 Lat, 25.628 Lon]. Kenya: Nkubu-Meru, m 1500, 13.xi.1988 [−0.065 Lat, 37.676 Lon]; Meru distr., Materi (Mitungu), 800 m, 2/15.xi.1988 [−0.161 Lat, 37.828 Lon]; S of Garissa, 15 kms S of Bura, 29.iv.2011 [−1.278 Lat, 39.947 Lon]; Kitwii [−1.314 Lat, 37.399 Lon]; S of Garissa, 10 kms S of Hola, 27.iv.2011 [−1.595 Lat, 40.024 Lon]; Kiboko, 21.xi.1999 [−2.191 Lat, 37.699 Lon]; 2 kms NW of Garsen, 14–17.xii.2009 [−2.260 Lat, 40.097 Lon]; Kenya Coast, 29 kms S of Garsen, 29.iv.2017 [−2.537 Lat, 40.123 Lon]; Mtito Andei, iv.1950 [−2.693 Lat, 38.161 Lon]; Amboseli N.P., Serena Lodge [−2.705 Lat, 37.268 Lon]; Tsavo N.P., Kilaguni Lodge, 800 m, 28.−30.XI.1991 [−2.906 Lat, 38.081 Lon] (from Flickr); Voi, Tsavo, 22.xi.−2.xii.1996 [−3.397 Lat, 38.552 Lon]; Taita Taveta, Wundanyi, 15.x.2019, −3.401 Lat, 38.365 Lon (from iNaturalist); Taveta, Tsavo West N.P., Lake Jipe, xii.1993 [−3.427 Lat, 37.667 Lon]; S of Voi, 23.xi.1997 [−3.427 Lat, 38.557 Lon]; ditto, 5.xii.2021; SW of Voi, 8–12.xii.2009 [−3.439 Lat, 38.530 Lon]; Rabai, xi-xii.1933 [−3.945 Lat, 39.559 Lon]; Mombasa, 10.vi.1974 [−4.043 Lat, 39.668 Lon]; Cheteni, xi.1911 [−4.105 Lat, 39.629 Lon]; Tiwi, xi.1911 [−4.240 Lat, 39.594 Lon]. Malawi: Karonga, Chilumba, 20.xii.2019, −10.439 Lat, 34.255 Lon (from iNaturalist); Salima env., 5–6.i.2002 [−13.783 Lat, 34.480 Lon]; Balaka env., 05–06.i.2002 [−14.995 Lat, 34.928 Lon]. Mali: Niger, Kouakourou [14.149 Lat, −4.457 Lon]; Koulouba, v-vi.1909 [12.664 Lat, −8.023 Lon]. Mozambique: Mamboia [−11.477 Lat, 35.931 Lon]; Ile d’Ibo [−12.345 Lat, 40.605 Lon]; Nyassa [−12.948 Lat, 32.291 Lon]; Lumbo, ii.1983 [−15.037 Lat, 40.670 Lon]; SW of Lake Chilwa [−15.581 Lat, 36.019 Lon]; Masiuene [−16.417 Lat, 39.833 Lon]; Zambesi, Boroma [−17.446 Lat, 35.66 Lon]; Vila Pery, 1928 [−19.143 Lat, 33.498 Lon]; Amatongas Forest, ii-iii.1962, D. Cookson, 1 ex. [−19.186 Lat, 33.774 Lon]; Sofala, 4.xii.2015 (from Flickr) [−19.821 Lat, 34.843 Lon]; Beira, 23.xii.1954 [−19.833 Lat, 34.871 Lon]; Rikatla [=Elias] [−24.059 Lat, 33.224 Lon]; Lourenço Marques [=Maputo] [−25.969 Lat, 32.573 Lon]; Maputo env., xi.1993 [−25.875 Lat, 32.342 Lon]; Delagoa Bay, x.1897 [−26.079 Lat, 32.559 Lon]. Namibia: Ovambo, Ohamwaala, −17.417 Lat, 16.105 Lon, 21.i.1993. Caprivi, Zambesi riv., Katima Mulilo, 15–24.i.1995 [−17.445 Lat, 24.310 Lon]. Nigeria: Samaru, 26–31.v.1970 [11.146 Lat, 7.628 Lon]. Republic of South Africa: Kruger N.P., Punda Maria, xi.2005 [−22.692 Lat, 31.017 Lon]; Waterberg Mountain, Bosveld, 31.xii.2018, −23.868 Lat, 27.700 Lon (from iNaturalist); Shiluvane [−24.039 Lat, 30.283 Lon]; Letapa, 29.xi.2019, −24.152 Lat, 30.593 Lon (from iNaturalist); Maruleng, 25.xi.2019, −24.333 Lat, 30.770 Lon (from iNaturalist); Ehlanzeni, Satara, 6.xii.2012, −24.476 Lat, 31.390 Lon (from iNaturalist); Mopani, 29.xii.2019, −24.523 Lat, 30.894 Lon (from iNaturalist); Ohrigstad env., 8.xi.1994 [−24.747 Lat, 30.579 Lon]; Ehlanzeni, Marloth Park, 12.xi.2018, −25.372 Lat, 31.764 Lon (from iNaturalist); Dinokana [Linokana] [−25.439 Lat, 25.828 Lon]; Barberton [−25.791 Lat, 31.028 Lon]; Elisras env., 831 m, 5.xi.2013 [−27.938 Lat, 23.806 Lon]; 22 kms SE of Kimberly, 1953 [−28.934 Lat, 24.869 Lon]; Congella, 1.vii.1926 [−29.876 Lat, 30.991 Lon]; Durban, 1902 [−29.858 Lat, 30.927 Lon]; Malvern [−29.886 Lat, 30.919 Lon]. Somalia: Mareerey, Kismaio, v.1983 [3.659 Lat, 43.452 Lon]; Gelib, Alessandra, iv.1937 [0.486 Lat, 42.783 Lon]. South Sudan: Equatoria [4.922 Lat, 30.897 Lon]; Nimule [3.587 Lat, 32.079 Lon]. Tanzania: Kilimandjaro, Rauya [−3.326 Lat, 37.540 Lon]; Kilimandjaro, Marangu, 1500 m [−3.269 Lat, 37.510 Lon]; Old Shinyanga [−3.681 Lat, 33.427 Lon]; West Usambara, Kwai [−4.751 Lat, 38.354 Lon]; West Usambara, Mombo [−4.885 Lat, 38.271 Lon]; Tanga Prov., 20–23.ii.1950 [−5.142 Lat, 39.038 Lon]; Segala Hills, xii.1993, Werner leg., 1 ex. [−5.579 Lat, 36.166 Lon]; Zanzibar [−6.162 Lat, 39.242 Lon]; Bagamoyo [−6.459 Lat, 38.919 Lon]; Pwani prov., 70 kms E Morogoro, −6.600 Lat, 38.133 Lon, 310 m, 11.xii.2006; Dar-es-Salaam [−6.780 Lat, 39.191 Lon]; Tabora, Sikonge, 12.xi.2021, Lat: −6.809 Lon: 33.251 (from iNaturalist); Morogoro, 3.vi.1986 [−6.837 Lat, 37.627 Lon]; Morogoro env. [−6.885 Lat, 37.636 Lon]; Ousagara [−7.314 Lat, 37.569 Lon]; 56 kms N Iringa, 26.i.2014 [−7.475 Lat, 35.738 Lon]; Rufiji River, leprosy centre near Utete, 1–3.xii.2014 [−7.985 Lat, 38.758 Lon]; 23 kms S Ifakara, 21–22.i.2014 [−8.355 Lat, 36.666 Lon]; Mbeya prov., 120 kms E Mbeya, −8.850 Lat, 34.000 Lon, 1220 m, 6.i.2007; Kigonsera [−10.795 Lat, 35.069 Lon]. Zambia: Mukolomba, 3500–400 ft., 27–30.x.1909 [−10.826 Lat, 29.036 Lon]; Kabwesha env., 3500–4500 ft, 24–30.x.1904 [−10.843 Lat, 28.921 Lon]; NW Province, Ikelenge, Kalene Hill, 1420 m, 7–8.xi.2015 [−11,188 Lat, 24.201 Lon]; Mubwa, 19.xii.2019, −14.973 Lat, 25.926 Lon (from iNatutalist); Mwengwa, −15.269 Lat, 26.038 Lon, xii.1913; Lusaka W, 25.v.88 [−15.349 Lat, 28.203 Lon]; Lusaka, Kafue City, Kafue River, m 1200, 22.xi/2.xii.1987 [−15.778 Lat, 28.150 Lon]; Gwembe, 23.i.2020, −16.543 Lat, 28.689 Lon (from iNaturalist). Zimbabwe: Victoria Falls, Zambesi N.P.-Camp, [−17.883 Lat, 25.816666 Lon]; Victoria Falls, xii.1915 [−17.940 Lat, 25.863 Lon]; Hwange N.P. [= Wankie], Main Camp nr. Pan 5 [−18.368 Lat, 26.518 Lon]; Mutare, Dorowa env., 29.xi.1998 [−19.057 Lat, 31.778 Lon]; Mutare, Bvumba env., 21–24.xii.1998 [−19.100 Lat, 32.783 Lon]; Mvuma, route Gutu-Chatsworth, 24.ii.1998 [−19.640 Lat, 30.878 Lon]; Lake Mutirikwe [−20.217 Lat, 31.000 Lon]; Matopos, −20.550 Lat, 28.500 Lon, 30.xi.1993; Matobo N.P., 3.v.1998, 50 kms S of Bulawayo [−20.576 Lat, 28.496 Lon]; Matébélé, Penda-Ma-Tenka [−20.597 Lat, 29.068 Lon].

*Polyclada pectinicornis* (Olivier)

*Clytra pectinicornis* Olivier, 1791: 31 [50].

*Diamphidia pectinicornis* (Olivier): Fairmaire 1887: 362 [51]; Gerstaecker 1873: 280–281 [52].

*Cladotelia pectinicornis* (Olivier): Kolbe 1897: 335 [43].

*Cladotelia pectinicornis* var. *nebulosa* Weise, 1902: 162 [44].

*Cladocera pectinicornis* (Olivier): Peringuey 1892: 88–89 (? var. *dispar*) [53]; Ferreira 1963: 520 [45].

*Polyclada pectinicornis* (Olivier): Bechyné 1955: 558 [54]; Bryant 1960: 352 [55]; Scherer 1961b: 174 [56].

*=Diamphidia compacta* Fairmaire, 1887: 361–362 [51] syn. nov.

*=Cladotelia compacta* (Fairmaire): Kolbe 1897: 335 [43].

Locus typicus: “Sénégal” (*P. pectinicornis*); “Makdishu” (Somalia) (*P. compacta*).

Type material: Lectotype ♂: “*Clythra Pectinicornis.ol*/Seneg./ex Musaeo G Olivier/Syntype/ Museum Paris/MNHN, Paris EC15854” (MNHN) (here designated by M. Biondi and P. D’Alessandro); paralectotypes 1 ♂ (EC15856) and 1 ♀ (EC15855), same data as lectotype (MNHN). Lectotype ♀: “*Diamphidia compacta* mihi/Makdischu/Ex-Musaeo L. Fairmaire 1893/MNHN, Paris EC12722” (MNHN) (here designated by M. Biondi and P. D’Alessandro).

Distribution: Angola (!), Benin (!), Chad, Congo (!), Ethiopia (!), Gambia (!), Ghana (!), Kenya, Malawi (!), Mali (!), Mozambique, Namibia (!), Niger (!), Nigeria (!), Republic of South Africa (!), Rwanda (!), Senegal, Somalia (!), Sudan (!), Tanzania, Zambia (!), and Zimbabwe (!). Chorotype: Afro-Intertropical (AIT).

Localities examined: Angola: 10 kms SE of Kinjama [−12.680 Lat, 21.359 Lon]. Benin: env. Porto Novo, Waterlot, 1912 [6.473 Lat, 2.569 Lon]. Chad: Fort Lamy [12.141 Lat, 15.027 Lon]; N’Gouri, Kanem distr. [13.638 Lat, 15.369 Lon]. Congo: 2 kms SE of Kingouala [−4.246 Lat, 14.509 Lon]. Ethiopia: Borana, Javello, xi-xii.1938/i-iii.1939 [4.883 Lat, 38.083 Lon]; near Arba Minch., Gemu Gofa prov., iv.1994 [6.024 Lat, 37.540 Lon]; El Rago, Ogaden, 15.xii.1952 [6.559 Lat, 45.786 Lon]; Aware Dagabuk, Duhun Area, 1953–54 [7.772 Lat, 43.541 Lon]; Sidamo, 18.iv.2007 (from Flickr) [7.850 Lat, 36.083 Lon]. Gambia: 2 kms NW of Bere Kobng [13.380 Lat, −15.327 Lon]. Ghana: Accra, xi.1965 [5.604 Lat, −0.187 Lon]. Kenya: Kiungani, x.1947 [0.942 Lat, 34.902 Lon]; Isiolo, iv.1956 [0.357 Lat, 37.607 Lon]; Meru distr., Gatunga, iv–v.1987 [−0.101 Lat, 38.008 Lon]; 32 kms E of Thika, Mwingi env., 30.iv.2011 [−1.033 Lat, 37.357 Lon]; Katutu-Kithioko, 27.xi.1999 [−1.127 Lat, 37.834 Lon]; Kiunga [−1.733 Lat, 41.517 Lon]; Kibwezi, 14.xii.1928 [−2.412 Lat, 37.968 Lon]; near Kibwezi, 2.xii.1996 [−2.423 Lat, 37.952 Lon]; Watita Hill, Kedai, xii.1911-i.1912 [−3.267 Lat, 38.364 Lon]; SW of Voi, 8–12.xii.2009 [−3.504 Lat, 38.453 Lon]; Taveta, Tsavo West N.P., Lake Jipe, xii.1993 [−3.599 Lat, 37.782 Lon]; Mombas [−4.069 Lat, 39.673 Lon]; Mombasa Island, Kilindini, 27.v−3.vi.1955 [−4.060 Lat, 39.651 Lon]. Malawi: 16 kms S of Sangala [−15.491 Lat, 35.112 Lon]. Mali: Sangha, Dogon Plateau, v.1992 [14.497 Lat, −3.493 Lon]. Mozambique: Valée du Pungoué, Guengère, 1906 [−19.083 Lat, 34.250 Lon]; Vila Pery [−19.143 Lat, 33.498 Lon]; Delagoy Bay [−26.079 Lat, 32.559 Lon]. Namibia: Otjozondjupa, Grootfontein, 16.i.2017, −19.505 Lat, 18.031 Lon (from iNaturalist). Niger: Agadès, entre L’Air et Niger, 1909 [16.991 Lat, 7.979 Lon]; Tibiri-Maradi (Mission Tilho), 1910 [13.541 Lat, 7.080 Lon]; Maradi, 19.vii.1980 [13.034 Lat, 7.089 Lon]. Nigeria: Kano distr., vi-vii.1953 [12.026 Lat, 8.606 Lon]; ditto, vii.1997; Bornu, Maiduguri, 1948–49 [11.754 Lat, 13.174 Lon]; Azare, 1926 [11.660 Lat, 10.236 Lon]; Ife, 9.iii.1972 [7.485 Lat, 4.560 Lon]. Republic of South Africa: Northern Province, near Ohrigstad, 8.xi.1994 [−24.749 Lat, 30.5799 Lon]. Rwanda: Lake Ihema, 25.xi.1971 [−1.841 Lat, 30.732 Lon]. Senegal: Saint Louis [16.034 Lat, −16.476 Lon]; 3 kms S of Tiaski Ndialal [15.572 Lat, −14.949 Lon]; Dakar, Senebgambien [14.695 Lat, −17.448 Lon]; 8 kms E of Toubéré Amadou [14.637 Lat, −13.199 Lon]; 3 kms SW of Boudou Mbaba [14.594 Lat, −14.586 Lon]; Somone, vii.1971 [14.487 Lat, −17.058 Lon]; Fatick, Diouroup, 8 m, 14–18.viii.2007 [14.365 Lat, −16.530 Lon]. Somalia: El Uach, Samarole, 3–4.xi.1986 [2.692 Lat, 41.451 Lon]; Bardera, Faafax-Dhuun/El Wak road [2.521 Lat, 41.446 Lon]; Afgoi, ii-iv.1978 [2.164 Lat, 45.037 Lon]; Mogadishu, iv-vi.1932 [2.064 Lat, 45.282 Lon]; Gelib, Alessandra, 1937 [0.491 Lat, 42.789 Lon]; Sar Uanle, ii-iii.1973 [−0.573 Lat, 42.322 Lon]. Sudan: Jebel Marra, Golal, 28.v.1963 [12.940 Lat, 24.356 Lon]; Darfur Province, Kulme, 5.vi.1921 [12.595 Lat, 23.619 Lon]. Tanzania: Mwanza region, Usagara [−2.681 Lat, 32.999 Lon]; Rombo, Lake Chala, 15.4.2021, −3.307 Lat, 37.689 Lon (from iNaturalist); Arusha, Naberera env., 8–13.iv.1997 [−4.204 Lat, 36.925 Lon]; Tabora, 3.500–4.000 ft, dry season [−5.064 Lat, 32.858 Lon]; Mlingano, Geigletz Estate, 4–12.iii.1950 [−5.116 Lat, 38.864 Lon]; Tanga, Ngomeni, Mlingano Sisal Research Stn., i-iii.1951 [−5.154 Lat, 38.899 Lon]; 8 kms SW of Tanga, iv.1912 [−5.142 Lat, 39.038 Lon]; Handeni, 350 m [−5.424 Lat, 38.1491 Lon]; Kakoma bei Tabora [−5.491 Lat, 32.626 Lon]; Segala Hills, xii.1993 [−5.577 Lat, 36.173 Lon]; Zanzibar [−6.095 Lat, 39.259 Lon]; Mhonda-Ouzigoua, 1879–1880 [−6.309 Lat, 37.549 Lon]; Bagamoyo [−6.459 Lat, 38.919 Lon]; Dar-es-Salaam [−6.770 Lat, 39.179 Lon]; Morogoro, 6.ii.1921 [−6.822 Lat, 37.638 Lon]; 10 kms NW of Magindu [−6.883 Lat, 38.389 Lon]; Pugu, 5.xii.1925 [−6.908 Lat, 39.143 Lon]; Uluguru Mts., iv.1997 [−7.109 Lat, 37.667 Lon]; 56 kms N Iringa, 26.i.2014 [−7.475 Lat, 35.738 Lon]; Utengule, 18.ii.2020 (from Flickr) [−8.890 Lat, 33.321 Lon]; 4 kms NE of Matema [−9.464 Lat, 34.049 Lon]. Zambia: Livingstone, Zambesi riv., 15.ii.1918 [−17.809 Lat, 25.826 Lon]. Zimbabwe: Victoria Falls, Zambesi riv. [−17.937 Lat, 25.805 Lon]; Dorowa env., Mutare, 29.xi.1998 [−19.007 Lat, 32.643 Lon]; Matabeleland, xii.1896 [−19.852 Lat, 28.081 Lon]. 

#### Morphological Remarks

*Polyclada bohemani* and *P. pectinicornis* are very similar in shape, proportions, and colour (Figure 1a and Figure 2a) and have therefore very often been confused with each other. This is mainly due to their wide intraspecific variability, especially in *P. pectinicornis*, but also to the fact that adequate studies on the morphology of their aedeagus and spermatheca had not been carried out. The two species, however, exhibit useful diagnostic characters relating to their external habitus and show some constant chromatic differences. Although major morphological differences relate to the median lobe of the aedeagus (Figure 1 and Figure 2), the two species can be reliably identified by focusing on the position of certain elytral patches: the hind sutural patches are generally aligned with the lateral patches in *P. pectinicornis* (Figure 2a), while in *P. bohemani* they are not aligned with the lateral patches and are placed closer to the elytral apex (Figure 1a). This latter species tends to be stable in colour over its distribution range, with black patches, antennae, and legs, except for hind femora which are mostly brownish. *P. pectinicornis* is instead more variable, with specimens from different localities having antennomeres 1–3 and scutella that are partially yellowish, along with lighter femora; specimens from East Africa often display brownish, and not black, elytral patches. 

With reference to the median lobe of the aedeagus, in *Polyclada bohemani* (Figure 1b) in the ventral view can be observed a sinuate shape that narrows in the distal fourth, an apical part that is medially very prominent, subromboidal, with a rounded median tooth; the ventral sulcus is visible in the apical third; in the lateral view can be seen a straight median lobe with the distal fourth clearly bent ventrally; dorsal ligula are elongated, moderately wide in basal 2/3 s and abruptly narrowed in the distal third; the apical part is bifid, dorsally curved and visible in the lateral view. In contrast, for *Polyclada pectinicornis* (Figure 2b) the median lobe of the aedeagus is fusiform in the ventral view, truncating apically, with a rounded median tooth; the distal surface is slightly depressed ventro-laterally, and medially prominent; in the lateral view, the median lobe is straight with the distal fourth slightly bent ventrally; dorsal ligula are elongate, wide but abruptly narrowed subapically; the apical part is formed by a wide medial lobe and two thinner lateral lobes, all dorsally bent and well visible in the lateral view. However, we observed some variability in the shape of the median lobe of the aedeagus, in particular in *P. bohemani*, where the specimens from more southern locations, for example KwaZulu-Natal, generally showed a more slender shape of the distal part. However, at present we prefer not to establish continuity solutions within this variability by the description of new taxa, pending future information deriving from phylogeographic assessment of these populations.

The spermatheca also displays some differences between the two species, although these differences are not always constant. In *Polyclada bohemani* (Figure 1c), the elongate basal part is generally slightly narrower towards the ductus, which is apically inserted, short and thickset; the distal part is thin, generally hook-shaped and apically rounded, with no distinct collum and no appendix. In *P. pectinicornis* (Figure 2c), the spermatheca is very similar but generally shows a subelliptical-elongate basal part and thin distal part, which is moderately curved and generally apically pointed.

Both species are also quite variable in size, although biometric ratios do not vary much within each species: 

*Polyclada bohemani*. Males (n = 176; range): 6.81 ≤ LE ≤ 9.56 mm; 4.38 ≤ WE ≤ 6.56 mm; 1.75 ≤ LP ≤ 2.50 mm; 3.38 ≤ WP ≤ 4.88 mm; 6.63 ≤ LAN ≤ 9.38 mm; 2.75 ≤ LAED ≤ 3.75 mm; 8.13 ≤ LB ≤ 12.69 mm; 3.83 ≤ LE/LP ≤ 3.89; 1.30 ≤ WE/WP ≤ 1.35; 1.93 ≤ WP/LP ≤ 1.95; 0.64 ≤ WE/LE ≤ 0.69; 0.74 ≤ LAN/LB ≤ 0.82; 2.48 ≤ LE/LAED ≤ 2.55; LA 100-42-67-108-117-108-117-108-117-108-258. Females (n = 142; range): 7.63 ≤ LE ≤ 11.38 mm; 5.88 ≤ WE ≤ 8.88 mm; 1.88 ≤ LP ≤ 2.75 mm; 3.78 ≤ WP ≤ 5.63 mm; 5.56 ≤ LAN ≤ 8.13 mm; 1.03 ≤ LSP ≤ 1.31 mm; 9.31 ≤ LB ≤ 13.75 mm; 4.07 ≤ LE/LP ≤ 4.14; 1.55 ≤ WE/WP ≤ 1.58; 2.02 ≤ WP/LP ≤ 2.05; 0.77 ≤ WE/LE ≤ 0.78; 0.59 ≤ LAN/LB ≤ 0.60; 7.39 ≤ LE/LSP ≤ 8.67; LA: 100-50-75-121-100-92-92-83-83-75-125.

*Polyclada pectinicornis*. Males (n = 143; range): 8.13 ≤ LE ≤ 9.63 mm; 5.06 ≤ WE ≤ 5.94 mm; 2.00 ≤ LP ≤ 2.50 mm; 3.88 ≤ WP ≤ 4.63 mm; 8.25 ≤ LAN ≤ 10.13 mm; 3.68 ≤ LAED ≤ 3.69 mm; 9.63 ≤ LB ≤ 12.88 mm; 3.85 ≤ LE/LP ≤ 4.06; 1.28 ≤ WE/WP ≤ 1.31; 1.85 ≤ WP/LP ≤ 1.94; 0.61 ≤ WE/LE ≤ 0.62; 0.79 ≤ LAN/LB ≤ 0.86; 2.20 ≤ LE/LAED ≤ 2.61; LA: 100-53-60-107-113-100-120-107-113-113-343. Females (n = 89; range): 9.63 ≤ LE ≤ 11.38 mm; 6.63 ≤ WE ≤ 9.00 mm; 2.50 ≤ LP ≤ 2.94 mm; 4.88 ≤ WP ≤ 5.94 mm; 6.75 ≤ LAN ≤ 8.13 mm; 1.24 ≤ LSP ≤ 1.25 mm; 11.38 ≤ LB ≤ 14.00 mm; 3.85 ≤ LE/LP ≤ 3.87; 1.36 ≤ WE/WP ≤ 1.52; 1.95 ≤ WP/LP ≤ 2.02; 0.69 ≤ WE/LE ≤ 0.79; 0.58 ≤ LAN/LB ≤ 0.59; 7.70 ≤ LE/LSP ≤ 9.10; LA: 100-47-67-100-87-80-87-77-87-80-153. 

### 3.2. Ecological Niche Modelling and Habitat Preferences

After the VIF and Pearson’s correlation analyses, we selected a set of nine uncorrelated bioclimatic variables (bio2: mean diurnal range, bio3: isothermality, bio8: mean temperature of the wettest quarter, bio9: mean temperature of the driest quarter, bio13: precipitation of the wettest month, bio14: precipitation of the driest month, bio15: precipitation seasonality, bio18: precipitation of the warmest quarter, and bio19: precipitation of the coldest quarter), which were then used to calibrate the models. 

The ensemble models for both the target species resulted in high performance scores (AUC = 0.946 and TSS = 0.766 for *P. bohemani*, AUC = 0.954 and TSS = 0.810 for *P. pectinicornis*), indicating a wide and continuous suitable area for *P. bohemani* (Figure 3a) in the eastern part of sub-Saharan Africa, and a more irregular and narrower suitable area for *P. pectinicornis* (Figure 3b). Concurrently, a sub-Saharan suitability band, spanning from Senegal to Sudan, was found to be suitable for *P. pectinicornis* only, in addition to a suitable patch in Chad, territories where *P. bohemani* scored low suitability values (Figure 3). 

When coupling this data with vegetation formations, a common Central African “core” is evident, along with a parallel division of sub-Saharan versus Southern African preference (Figure 4). 

Indeed, this translates into a higher preference of *P. bohemani* for denser savanna or forest environments, such as tropical lowland grassland savanna and shrubland (2.A.1) and tropical seasonally dry forest (1.A.1), while *P. pectinicornis* is more associated with dryer environments, such as warm desert and semi-desert scrub and grassland (3.A.2) and salt marsh (2.B.7) formations (Figure 5). 

This trend is also confirmed by the tree cover results, where *P. bohemani* showed a higher affinity to greater tree cover values (Figure 6a), while *P. pectinicornis* was found to prefer more open habitats (Figure 6b).

### 3.3. Environmental Niche Overlap

Within the PCA-Env calibrated using the same set of variables as the ENMs, together with percentage cover of the vegetation formations 1.A.2, 2.A.1 and 3.A.2, the first two principal components (PrinComp1 and PrinComp2) explained 56.8% of overall environmental variability across sub-Saharan Africa (Figure 7a). Percent cover of warm desert and semi-desert scrub and grassland (3.A.2) was strongly associated with the positive semi-axes of both the PrinComps, together with mean temperature of the wettest quarter (bio8) and precipitation seasonality (bio15). Meanwhile, percentage cover of tropical lowland humid forest (1.A.2) contributed to the negative semi-axis of PrinComp1, along with isothermality (bio3) and precipitation of the driest month (bio14). In contrast, percentage cover of tropical lowland grassland, savanna and shrubland (2.A.1) was mainly associated with the negative semiaxis of PrinComp2. In the resulting 2-D environmental space (Figure 7b), *Polyclada bohemani* and *P. pectinicornis* showed kernel-smoothed densities of occurrence which almost overlapped, particularly in their core zones (i.e., those with highest ‘niche occupancy’ values). However, *P. pectinicornis* niche space extended more than that of *P. bohemani* towards the positive half of PrinComp1 and PrinComp2 (Figure 7b), thus suggesting that the former species responds positively to higher temperatures during the wettest period and to more extensive cover by xeric vegetation formations. Nonetheless, niche overlap between the two species was quite high when measured with Schoener’s *D* (*D* = 0.65), and even more so when using the modified Hellinger’s distance metric (*I* = 0.78). Based on the performed niche similarity tests, the alternative hypothesis of niche divergence between the two species could be confidently rejected (*p* = 0.988), while that of niche conservatism received quite strong support (*p* = 0.02).

## 4. Discussion

*Polyclada bohemani* and *P. pectinicornis* are well characterized from a morphological point of view, in particular as regards the shape of the aedeagus. However, this shows some small differences at the local level that should be better evaluated in the future with the support of molecular assessment. The wide chromatic variability of these two flea beetle species, especially in *P. pectinicornis*, has often been a cause of confusion in their identification, and until now has prevented correct definition of their distribution in the Afrotropical region. *P. bohemani* and *P. pectinicornis* are widely distributed in sub-Saharan Africa, with a large overlapping area in East Africa particularly in Kenya and Tanzania. However, while *P. bohemani* was found to have areas of high suitability in the more south-eastern regions, including Mozambique, Zimbabwe, and the eastern part of the Republic of South Africa, the areas with higher values of suitability for *P. pectinicornis* are located in the more northern regions, in particular in the belt close to the Sahara. This implies that both species, while showing preferences for savannah and open forest habitats, maintain certain differences in their choice of habitat. *P. pectinicornis*, in fact, is more frequently present than *P. bohemani* in sub-desert environments, while the latter can tolerate environments with dense tree cover. That is also confirmed by the environmental niche overlap analysis, which suggests that *P. pectinicornis* responds more positively than *P. bohemani* to higher temperatures during the wettest periods, as well as to more extensive xeric vegetation. 

## Figures and Tables

**Figure 1 insects-13-00668-f001:**
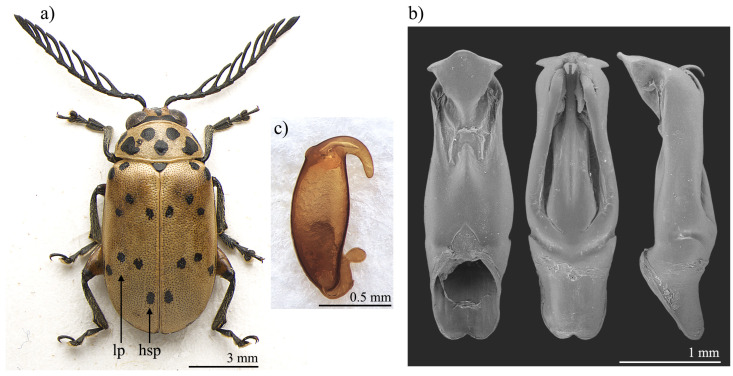
*Polyclada bohemani*: (**a**) habitus; (**b**) aedeagus from left to right in ventral, dorsal, and lateral view; (**c**) spermatheca. hsp: hind sutural patch; lp: lateral patches.

**Figure 2 insects-13-00668-f002:**
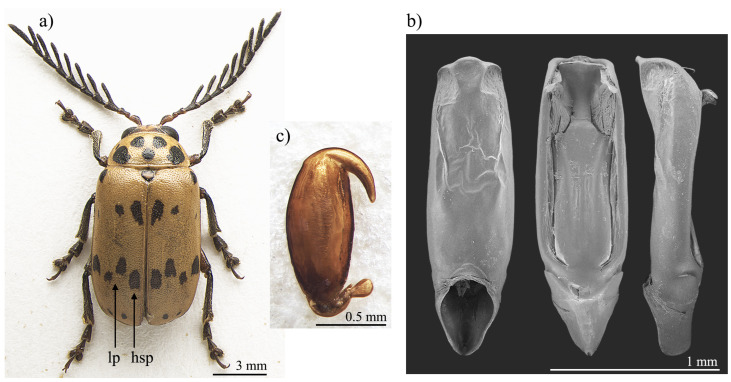
*Polyclada pectinicornis*: (**a**) habitus; (**b**) aedeagus from left to right in ventral, dorsal, and lateral view; (**c**) spermatheca. hsp: hind sutural patch; lp: lateral patches.

**Figure 3 insects-13-00668-f003:**
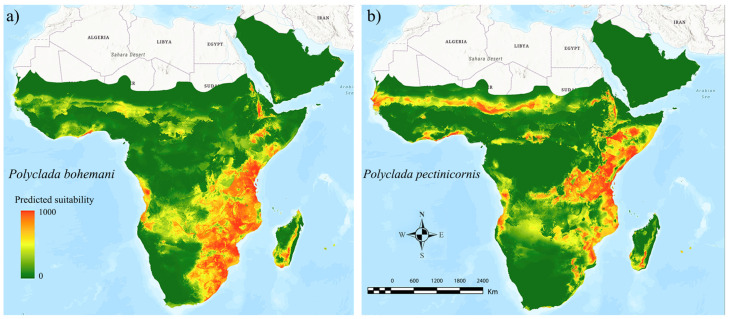
Ecological niche models inferred for current climatic conditions for (**a**) *Polyclada bohemani* and (**b**) *Polyclada pectinicornis* in sub-Saharan Africa. The green-to-red scale indicates low-to-high predicted suitability.

**Figure 4 insects-13-00668-f004:**
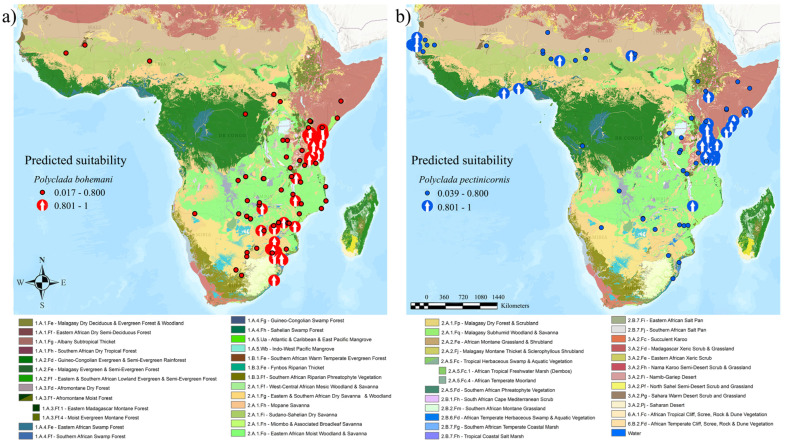
(**a**) *Polyclada bohemani* and (**b**) *Polyclada pectinicornis* localities, exceeding (arrows) or not (dots) 80% predicted climatic suitability, and their occurrence in the context of African vegetation formations.

**Figure 5 insects-13-00668-f005:**
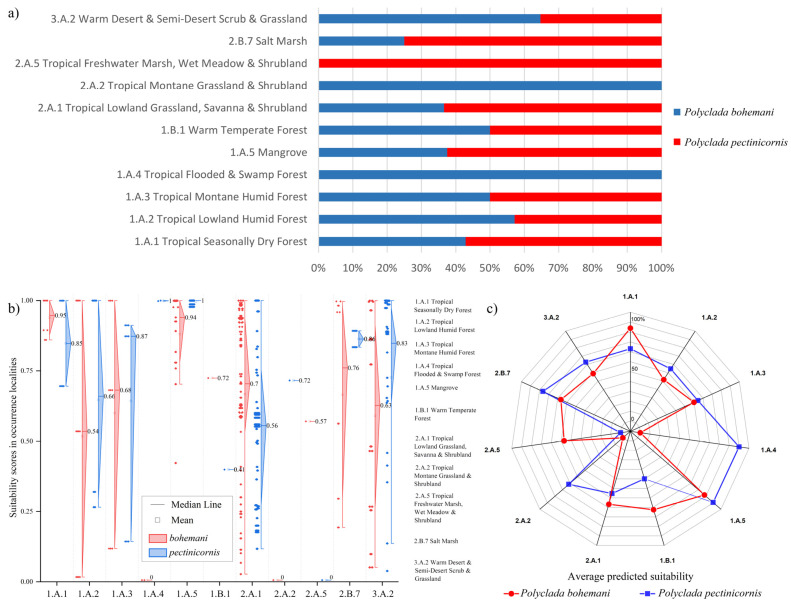
(**a**) Percent of localities of *Polyclada bohemani* and *Polyclada pectinicornis* falling within the reported vegetation formations; detailed suitability scores presented as (**b**) diamond boxplots with the corresponding localities’ vegetation formations, and (**c**) an averaged suitability for each vegetation formation.

**Figure 6 insects-13-00668-f006:**
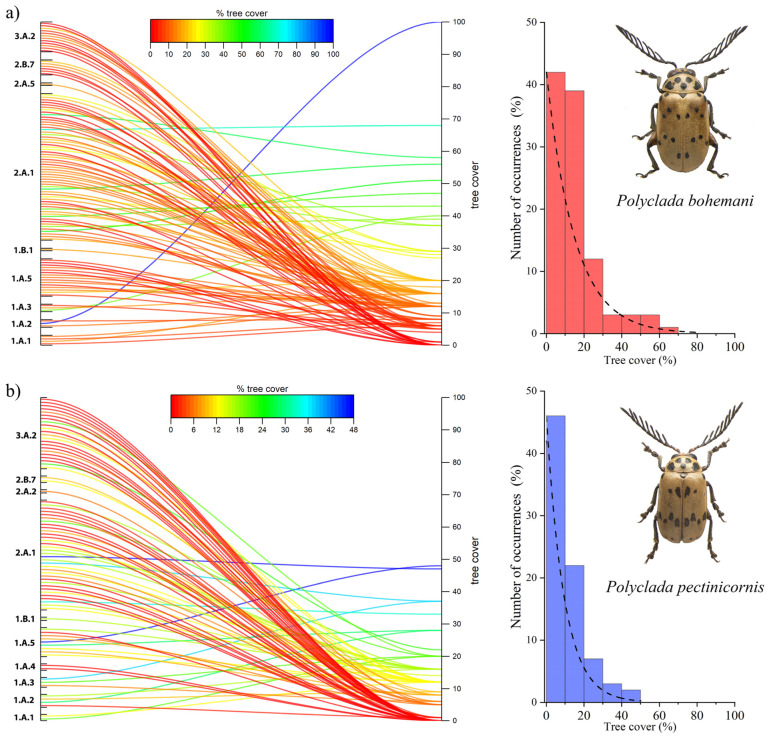
Percentage of tree cover and vegetation formation in localities (left), and histograms and corresponding fit for the number of occurrences (%) in relation to tree cover, for (**a**) *Polyclada bohemani* and (**b**) *Polyclada pectinicornis*.

**Figure 7 insects-13-00668-f007:**
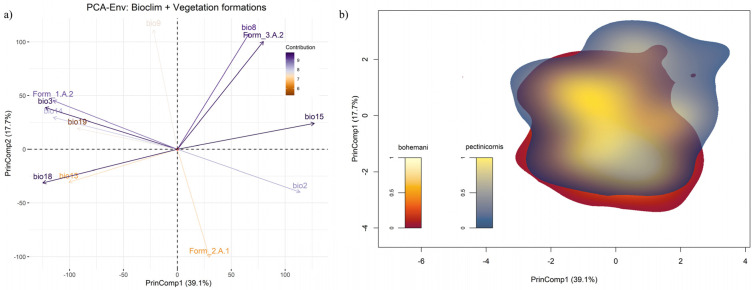
(**a**) Principal component analysis plot resulting from PCA-Env performed over the selected bioclimatic variables and vegetation formations; (**b**) two-dimensional, kernel-density-based space resulting from the Schoener’s and modified Hellinger’s distance-based analysis (performed over the variables mentioned previously), showing the niche overlap between *Polyclada bohemani* and *Polyclada pectinicornis*.

## Data Availability

Data other than the ones presented in this paper or information can be requested to the authors.

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
