# Peer review of "Taxonomy, Habitat Preference, and Niche Overlap of Two Arrow-Poison Flea Beetle Species of the Genus Polyclada in Sub-Saharan Africa (Coleoptera, Chrysomelidae)"

_insects, 2022, doi:10.3390/insects13080668_

Round 1
Reviewer 1 Report
The morphological illustrations and detailed measurement data are excellent and demonstrate how difficult it is to morphologically differentiate these two species. It appears that really only males can be differentiated using the aedeagal morphology. The elytral pattern differences are also apparent, but less so.
Based on the literature, the primary arrow/spear poison flea beetle species used by the San Bushman belong to the related genus Diamphidia that feed on Comiphora africana and, secondarily (i.e., less commonly used) the species of Polyclada feed on Sclerocarya caffra. Interestingly, neither of these plants are themselves toxic! However, the only mention in this manuscript of the host plants of these species of Polyclada is that they are “associated” with Anacardiaceae and Burseraceae. Most flea beetles are either oligophagous or monophagous. In the case of Polyclada I believe their tendency is more monophagous. So, aren’t the exact “associations” (i. e. specific hostplants for each species known), especially given how much attention and publications their study has produced?? I find it a little strange that the authors in this paper did not discuss at all the food plants or their distribution. Afterall it is the foodplant phytogeography, ecology, and distribution that dictates where these beetles will be found. Even in the “Localities Examined” sections in this manuscript that contain a lot of detailed data, there was no listing or mention of the valuable foodplant data that must have accompanied the other data!!
The statistical analysis and eco-modelling in this paper based on temperature, precipitation, vegetation formations provide interesting and useful ideas for their distribution, but these are herbivores, and it is their foodplant species that are the most important factors.
Also, while the distributions of these two species of Polyclada do overlap the San Bushman territory, much or even most of their distribution is further east and north.
Admittedly I am not a statistician. The statistical analyses in this manuscript seem excellent and very well-illustrated with resulting graphics. The authors have a long history of very nice statistical analysis of flea beetles. However, as mentioned above I still find it awkward that there is no discussion of the food/host plants.
Author Response
Dear Reviewer,
Thank you for your comments and suggestions.
The main issue you find is related to the host plants the Polyclada are associated with. Unluckily, the only reliable data for the two target species are the ones already reported in the Introduction (L 60-62); they are general information but represent the only known evidence. In fact, none of the examined materials we used (despite our database size) reports information on the host plants. Therefore, we could not mention any in the “Localities examined” section.
Of course, if we had some robust information, we would have added to the manuscript the host plants for each species and treated them statistically, for instance, as we did in our recent paper “Habitat Specificity, Host Plants and Areas of Endemism for the Genera-Group Blepharida s.l. in the Afrotropical Region (Coleoptera, Chrysomelidae, Galerucinae, Alticini)” - Insects 2021, 12, 299. https://doi.org/10.3390/insects12040299.
Reviewer 2 Report
see attached file

Author Response
The lectotypes were designated, as suggested by the reviewer (please, see the revised MS).
Regarding “Line 272 var. nebulosa”: This information is already reported in the manuscript (Line 274 of the revised MS): “Cladotelia pectinicornis var. nebulosa Weise, 1902: 162 [44]”. In fact, var. nebulosa was described by Weise in 1902.